# A biofeedback-enhanced therapeutic exercise video game intervention for young people with cerebral palsy: A randomized single-case experimental design feasibility study

Alexander MacIntosh[1,2,3]*, Eric Desailly[4☯], Nicolas Vignais[3,5☯], Vincent Vigneron[6☯], Elaine Biddiss[1,2☯]

**1** Bloorview Research Institute, Holland Bloorview Kids Rehabilitation Hospital, Toronto, Canada, **2** Institute of Biomaterials and Biomedical Engineering, University of Toronto, Toronto, Canada, **3** Complexité, Innovation, Activités Motrices et Sportives, Sciences du Sport, de la Motricité et du Mouvement Humain, Université Paris-Saclay, Orsay, France, **4** Recherche et innovation, Fondation Ellen Poidatz, Saint Fargeau-Ponthierry, France, **5** Complexité, Innovation, Activités Motrices et Sportives, Université d'Orléans, Orléans, France, **6** Informatique, Bio-informatique et Systèmes Complexes, l'Université d'Evry Val-d'Essonne, Evry, France

☯ These authors contributed equally to this work.
* alexander.macintosh@mail.utoronto.ca

**Data Availability Statement:** All relevant data are within the paper and its Supporting Information files.

## Abstract

### Importance/Background

Movement-controlled video games have potential to promote home-based practice of therapy activities. The success of therapy gaming interventions depends on the quality of the technology used *and* the presence of effective support structures.

### Aim

This study assesses the feasibility of a novel intervention that combines a co-created gaming technology integrating evidence-based biofeedback and solution-focused coaching (SFC) strategies to support therapy engagement and efficacy at home.

### Methods

Following feasibility and single-case reporting standards (CONSORT and SCRIBE), this was a non-blind, randomized, multiple-baseline, AB, design. Nineteen (19) young people with cerebral palsy (8–18 years old) completed the 4-week home-based intervention in France and Canada.

Participant motivations, personalized practice goals, and relevance of the intervention to daily activities were discussed in a Solution Focused Coaching-style conversation pre-, post-intervention and during weekly check-ins. Participants controlled a video game by completing therapeutic gestures (wrist extension, pinching) detected via electromyography and inertial sensors on the forearm (Myo Armband and custom software).

*Process feasibility* success criteria for recruitment response, completion and adherence rates, and frequency of technical issues were established *a priori*. *Scientific feasibility*, *effect*

**Funding:** This work was supported by the Canadian Institutes of Health Research [funding reference number RN304779 - 379428, Author AM], the Campus France Eiffel Excellence Scholarship [895191H, Author AM], and the Mitacs GobaLink program [IT10648, Author AM]. The funders had no role in study design, data collection and analysis, decision to publish, or preparation of the manuscript.

**Competing interests:** The authors have declared that no competing interests exist.

*size estimates and variance were determined for Body Function* outcome measures: active wrist extension, grip strength and Box and Blocks Test; and for *Activities and Participation* measures: Assisting Hand Assessment (AHA), Canadian Occupational Performance Measure (COPM) and Self-Reported Experiences of Activity Settings (SEAS).

## Results

Recruitment response (31%) and assessment completion (84%) rates were good and 74% of participants reached self-identified practice goals. As 17% of technical issues required external support to resolve, the intervention was graded as *feasible with modifications.* No adverse events were reported.

Moderate effects were observed in Body Function measures (active wrist extension: SMD = 1.82, 95%CI = 0.85–2.78; Grip Strength: SMD = 0.63, 95%CI = 0.65–1.91; Box and Blocks: Hedge's g = 0.58, 95%CI = -0.11–1.27) and small-moderate effects in Activities and Participation measures (AHA: Hedge's g = 0.29, 95%CI = -0.39–0.97, COPM: r = 0.60, 95% CI = 0.13–0.82, SEAS: r = 0.24, 95%CI = -0.25–0.61).

## Conclusion

A definitive RCT to investigate the effectiveness of this novel intervention is warranted. Combining SFC-style coaching with high-quality biofeedback may positively engage youth in home rehabilitation to complement traditional therapy.

## Trial registration

ClinicalTrials.gov, U.S. National Library of Medicine: NCT03677193.

## Introduction

Interactive computer play (ICP) is *"any kind of computer game or virtual reality technology where the individual can interact and play with virtual objects in a computer-generated environment"* [1]. It is an attractive way to augment traditional therapy and align with children's interest. Eight of 10 young people with Cerebral Palsy (CP) enjoy playing video games recreationally [2]. Cerebral Palsy (CP) is neuromuscular disability impacting approximately 2.11 per 1,000 live births in high-resource settings [3]. CP presents with positive and negative motor signs including: spasticity, weakness, impaired selective motor control and sensory deficits [4]. ICP has been used to improve balance [5], gait symmetry [6], upper limb strength [7] and other functional abilities in people with CP. Perhaps the largest study to date in this field is the 'Move it to improve it' (MITII) randomized controlled trial that used web-based therapy at home to improve occupational performance and visual perception in children with unilateral CP [8]. These types of studies have shown moderate evidence towards improving balance and overall motor skill but weak evidence towards improving upper extremity skills, joint control, gait and strength [9]. In part, the success of ICP-based therapies in the home has been thwarted by the challenge of (i) delivering high quality feedback that informs individuals with CP on any progress towards therapy goals as a therapist would in a clinic setting [10]; and (ii) sustaining engagement in the intervention over an extended period. Children with CP and their

families have expressed a desire for more accurate feedback within ICP [8] and reported the challenge of sustaining intrinsic interest in ICP therapies in the home.

## High quality feedback

Improving feedback quality in ICP can increase practice engagement and efficiency at home [11]. Biofeedback, where a people receives information about their body state (*e.g.* heart rate, foot speed, muscle activity) [1], can help increase awareness and control by informing them how their body is functioning [11]. Echoing families' remarks, a recent systematic review found that most interventions using biofeedback do so in a way that positions the people in a passive role in their practice and builds dependency, hindering motor learning progress [10].

Working with young people with CP and clinicians, we recently co-created an evidence-based biofeedback strategy (S1 File). The implementation improves biofeedback use (efficiency, effectiveness, engagement) by offering autonomy, varying feedback presentation (e.g. visual, audio), and being proportionate to the person's ability [10]. These mechanisms serve to improve the effectiveness of ICP therapies. They help direct the person to higher-quality movements at home where practice is completed without therapist supervision and provide a stronger cognitive link between game-focused (*e.g.* scoring points) and therapy-focused (*e.g.* decreased compensatory movements) goals (S1 File).

## Sustaining engagement

Low adherence is a primary concern in home-based interventions, historically ranging from 34–67% [12,13]. A recent review consistently found engagement and adherence difficult to maintain [14]. To improve adherence and engagement, an intervention must closely align with participants values and sustain their intrinsic motivation [14]. Intrinsic motivation is influenced by both the ICP technology (feedback, activity personalization), and the intervention design (therapist interaction, social support). Solution-focused coaching in pediatric rehabilitation (SFC-Peds) is a model of coaching recommended for youth with disabilities [15]. SFC-Peds builds intrinsic motivation to generate personal interest in health behaviour changes [16,17]. In SFC-Peds, coaches collaborate with children to help them envision their "preferred future" [15]. Through this process, children develop therapy goals and a supporting plan that aligns with their priorities.

## Aim

The success of an ICP intervention is influenced both by the technology used and the supports provided. In this project, we investigate the combination of a novel ICP technology integrating evidence-based biofeedback and Solution Focused Coaching strategies to promote home-based practice of hand/arm exercises. The ICP is a video game where participants complete therapeutic hand gestures to control game actions on-screen. The approach aims to provide a motivational, goal-based environment to address muscle weakness and selective motor control. This paper addresses intervention feasibility as articulated by Thabane et al (2010) [18]. This framework highlights that the aim of feasibility testing can be related to one or more of the following four classifications: process, resources, management and scientific. The objective here is to support development towards future randomized controlled trials and not to define statistical or clinical effectiveness of the intervention [18]. In this study, we concentrate on two of the four feasibility classifications:

First, we assess *process feasibility* of the biofeedback-enhanced therapy video game intervention protocol for young people with CP. The objective here is to determine the ability to enroll participants, enable home-based practice, and retain their activity during a 1-month

intervention. To this purpose, *a priori* success criteria were established for the recruitment and response rates, adherence, and frequency of technical difficulties impeding home practice [18].

Second, we assess the *scientific feasibility* of the intervention by estimating the effect size and variance for six person-centred outcome measures for the hand and wrist. The measures are aligned to the Body Functions and Activities and Participation chapters of the ICF (international classification of functioning disability and health) [19,20].

## Methods

### Design

A randomized, multiple-baseline, single-case experimental design (SCED) with two phases was applied. SCED designs can provide strong evidence wherein participants serve as their own controls for the purpose of within-subject comparison [21]. SCED research typically involves collecting a representative baseline phase through repeated measurements of an outcome of interest (Phase 1) that is then compared with the intervention phase (Phase 2). In a randomized SCED design, the time lapse between Phase 1 and Phase 2 is randomly allocated for each participant to further mitigate threats to internal validity [22]. SCED designs are increasingly used in clinical intervention research, particularly when sample size is limited, SCEDs can provide a rigorous approach to generate higher quality evidence [23]. In this feasibility study, the Single-Case Reporting Guideline in Behavioural Interventions (SCRIBE) [21] and the Consolidated Standards of Reporting Trials (CONSORT) methodologies were used (see S2 File and S3 File) [24]. Participants, researchers and assessing therapists were not blind to treatment phase. One methodological change occurred during the first week of enrollment. Inclusion criteria was expanded to include those with mixed tone and mild dystonia since it was found that they could control the game effectively and the potential therapeutic value was confirmed by clinicians. Procedural fidelity was maintained by following a standard operating protocol outlining the activities and resources for each phase (S4 File). Ethical approval was obtained by Holland Bloorview's Research Ethics Board (approved July 27, 2018, amended November 23, 2018, to include participants with mixed tone and mild dystonia) and the French national Comité de protection des personnes (CPP) (approved July 6, 2018). The authors confirm that all ongoing and related trials for this intervention are registered.

### Participants

From September to October 2018, ten participants were recruited from a regional rehabilitation hospital at a metropolitan city in Canada. From November 2018 to January 2019, ten participants were recruited from a rural regional rehabilitation hospital in France.

Inclusion criteria:

1. Cerebral Palsy diagnosis.

2. 8–18 years old.

3. Manual Abilities Classification System levels I-III [25].

4. Having a goal relating to improving hand/wrist function.

5. Dominantly spastic presentation. This original criterion was expanded to included mixed tone and mild dystonia.

6. Normal or corrected-to-normal vision and hearing.

7. Able to co-operate, understand, and follow simple instructions for game play.

8. Having passive wrist extension at least 10˚ greater than active wrist extension.

   Exclusion criteria:

1. Receiving active therapy of the hand/wrist.

2. History of unmanaged epilepsy.

3. Having received a Botulinum Toxin treatment within 3 months or constraint-based movement therapy within 6 months before study enrollment.

4. Visual, cognitive or auditory disability that would interfere with play.

5. Unable to commit an estimated minimum of 5-hours to training plan over four weeks.

**Sample size rationale.**   The sample size of twenty was determined based on recommended samples of a future definitive RCT with a small (0.2)–medium (0.5) effect, 90% power and two-sided 5% significance. Based on estimates of previous interventions on active wrist extension in the literature [26–29] a pre/post change of 10˚ with an expected standard deviation of 4–12˚ would require between 8–22 participants [30].

**Recruitment.**   Occupational therapists and developmental pediatricians identified participants who met inclusion criteria from their existing or previous caseload. In Canada, eligible participants were also identified through the hospital's centralized recruitment database, connect2research. In Canada the researcher telephoned potential participants after sending an invitation letter by mail. Then, the researcher screened interested participants and obtained written informed consent. Recruitment in Canada took place from September-November 2018. In France, the developmental pediatricians invited eligible individuals to participate. Developmental pediatricians obtained written informed consent from interested participants. Recruitment in France took place from November 2018—January 2019. Caregivers gave consent and were consulted to ensure the child could provide consent or assent. If capable, the child gave consent/assent [31].

## Protocol

The description follows the Template for Intervention Description and Replication (TIDieR guidelines), see S5 File [32]. S6 File.

**Baseline (phase A).**   Participants met once with the researcher and Occupational Therapist (60 minutes) in clinic. In a Solution-focused Coaching style [15,17] conversation, they discussed: motivations for participating, personalized scheduling and practice goals, how the intervention connects to daily activities. By the end of this conversation, participants established Canadian Occupational Performance Measure (COPM) goals [33]. Caregivers were present if desired. The dialogue was intended to improve cognitive engagement and consequently, home-play adherence [16]. Therapists and researchers guiding these conversations received one day of formal training and practiced scenarios with a certified SFC coach. Fidelity to the coaching style was maintained by following a conversation checklist developed with the certified SFC coach (S4 File).

Following the coaching conversation, a therapist assessed bimanual performance (Assisting Hand Assessment (AHA)) and gross manual dexterity (Box and Blocks (B&B)). The researcher visited each participant's home for multiple baseline testing of wrist extension and grip strength (3–6 visits, 30-minutes sessions). The number of baseline sessions was 'data-driven' to establish stability in the primary measure of effectiveness: active wrist extension–open fingers (AWEO). Stability was defined as 80% of phase 1 data within interquartile range [34].

After baseline, participants waited a computer-generated randomized number of days (between 1–10 days) to begin the intervention.

**Activity description.** During 1–2 baseline sessions the researcher habituated the participant to ICP activity system controls. Participants learned to control the video game using a therapeutic gesture, one of: wrist extension- open fingers, wrist extension- closed fingers, finger-thumb pinch, supination. In the SFC-Peds style conversation, therapists helped participants identify which gesture to practice based on the daily activities that were established to be important to them. All but two participants practiced wrist extension. In the game, participants are rewarded by making the gesture at the correct time with high quality (*i.e.* high forearm extensor activity and isolated hand movement).

**Intervention (phase B).** After the randomized waiting period, the researcher gave participants the system to practice at home. The system includes hardware: laptop, electromyography (EMG) and inertial sensor (Myo Armband) and software: adapted commercial video game (Dashy Square) and custom software to interpret movements and control the game (MATLAB 2017b). Participants practiced at home alone for 4-weeks according to their self-defined practice schedule established during the initial conversation. Once per week, the researcher visited each participant. During the 60-minute visit they:

a. Recorded gameplay with a video camera and electro-goniometer

b. Measured wrist extension and grip strength

c. Had a 'check-in' conversation to re-evaluate the self-defined practice goals

The check-in followed the SFC conversation guideline, EARS (elicit, amplify, reinforce, start again) [35] and served to gauge: satisfaction with progress, motivation, and modify practice goals and game difficulty if necessary. At the first and last weekly visit, participants completed Self-Reported Experiences of Activity Settings (SEAS) questionnaire to measure interaction, engagement, and sense of control while playing at home [36].

## Post-intervention

Within 2-weeks following the final visit, participants returned to clinic for a 60-minute assessment with the researcher and occupational therapist. The therapist re-evaluated: bimanual performance (AHA), gross manual dexterity (B&B), and COPM goals. The researcher conducted a semi-structured interview to gain participant's subjective evaluations of the intervention [37] (S4 File). Finally, a separate member of the research team (other than who completed the home visits) made a (5–10 minute) telephone call with participant's caregiver. The researcher asked questions and noted responses related to system use and integration into home-life (S4 File).

## Outcomes

To address *process feasibility*, *a priori* success criteria were compared to observed outcomes. Success criteria were based on previous recruitment, adherence and technical performance results of a home-based exercise intervention in a similar population [38,39]:

Process feasibility, a priori success criteria:

1. ≥10% response rate from all eligible participants [38]

2. ≥80% of the participants successfully complete the study. (i.e. completed at least 3 repeated measures during phase A and B, and complete assessments at baseline and post-intervention)

3. Participants meet their self-identified practice goal. (within ≥66% of the identified frequency and duration)

4. Participants were not prohibited from practicing due to technical constraints (*e.g.* After instruction, participants could start and play the game, technical challenges were overcome with the provided aid, and participants were not forced to cancel a practice session due to technical limitations). Participants may report multiple issues during the 1-month intervention.

The study is then given one of the following recommendations: Not feasible, Feasible with minor modifications, Feasible with close monitoring, Feasible as is according to criteria set by [18].

Towards assessing *scientific feasibility*, the size and variance of the effect of the intervention on participant-centred outcome measures are evaluated. As no single measure covers all aspects of function and experience during home-based interventions, complementary measures are used to capture changes across two ICF chapters [19,20].

1. Measures for ICF chapter: Body Function, changes in:

   a. active wrist extension–open fingers (AWEO)

   b. grip strength (GRIP)

   c. gross manual dexterity (B&B)

First, to evaluate the capacity with which participants can open their hand, a manual goniometric measurement of **active wrist extension** [40] with open and closed fingers was made. Participants start with the elbow in 90 degrees of flexion, the forearm pronated to the extent possible and the upper arm alongside the trunk. With the forearm fixed by the assessor, the child performs three wrist extensions per side [41]. Positive values indicate extension above neutral wrist position and values are recorded to the nearest five degrees, the minimal detectable difference [42]. In the relevant population passive movement tests show very good test–retest reliability (Intraclass correlation coefficients (ICC): 0.81–0.94) and moderate inter-rater reliability (correlation coefficients between 0.48–0.73 [43]). Second, **grip strength** was measured using a modified sphygmomanometer to evaluate relative changes in grip capacity [44]. The child sits with the arm adducted, the elbow flexed at 90 degrees and the forearm and wrist in neutral position (if possible). Participants maximally squeeze the device three times per side. The test is completed on both sides and the relative strength of the non-dominant hand to the dominant is reported. While normative data for children's grip strength using the modified sphygmomanometer are not available, [44] found excellent test-retest reliability (Pearson correlation coefficient of 0.97) and [45] reported high intra-rater reliability (ICC = 0.92). Third, the **Box and Blocks Test (B&B)** measured gross manual dexterity. The number of blocks a participant can move over a center divider in one minute is counted for both hands [46]. The B&B test shows high inter-rater reliability (ICCs >0.95) and test-retest reliability (ICCs >0.95) in children 6–19 years [47] and in adults with hemiplegia [48].

2. Measures for ICF Chapter: Activities and Participation, changes in:

   a. functional bimanual performance (AHA)

   b. perceived functional performance in a self-identified goal (COPM)

   c. perception of meaningful participation experiences (SEAS)

Three measures address ICF chapter, Activities and Participation. First, the ***Assisting Hand Assessment (AHA)*** quantifies spontaneous functional bimanual performance. Progressing through a board game guided by a trained occupational therapist, participants complete bimanual tasks such as opening a box or shuffling cards [49]. There are twenty tasks, scored on a 4-point scale. The smallest detectable change for the AHA is 5 logit units (scaled from 0–100). In adolescents with unilateral CP up to age 18, AHA shows good construct validity and excellent inter-rater (ICCs 0.94–0.98) and test-retest reliability (ICCs 0.98–0.99) [49,50]. Second, ***Canadian Occupational Performance Measure (COPM)*** evaluates perceived changes in satisfaction and performance of self-identified goal areas [33]. In conversation with a trained Therapist, participants rated a primary, and secondary if desired, goal area(s) from 1–10 in terms of importance, performance and satisfaction. Goal areas were re-evaluated post-intervention [33]. The primary goal's perceived change in performance and satisfaction are evaluated here [51]. 2-points is the minimal clinically meaningful change [52]. COPM has been reported as valid, reliable, and responsive [53]. Third, the ***Self-Reported Experiences of Activity Settings (SEAS) questionnaire*** evaluates experience with an activity in four domains: personal growth, psychological engagement, meaningful interactions, choice and control [36]. It is a 22-item questionnaire completed independently or with parental/researcher assistance. Scores are interpreted on a 7-point Likert scale (from +3- -3). The questionnaire has good internal consistency and test-retest reliability (Cronbach's alpha from 0.71 to 0.88, mean scale ICC = 0.68) [54].

## Analyses

For *process feasibility*, participant recruitment and demographic characteristics are presented as per CONSORT recommendations [24]. Success criteria are reported descriptively and narratively. Technical issues and resources required for resolution were documented and reported. Recommendations for the design of a future clinical trial is based on the number of and extent to which success criteria were met.

For *scientific feasibility* related to Body Function measures size and variance of the intervention effect on active wrist extension and grip strength are calculated for baseline and intervention phases. SCRIBE recommends a combined visual and statistical approach for SCED data [55]. The statistical approaches improve the estimate of the effect and can help account for serial dependence, variability and trends in the time-series data [23,56]. As the objective of the study is to inform a definitive RCT, the analysis focuses on estimating treatment effect size and variance. Statistical significance tests are not recommended to report due to insignificant power [21,57,58]. Level- and slope-change differences between phases, percentage of all non-overlapping data (PAND), and standardized mean difference (SMD, d-statistic with 95% confidence interval (CI)) show the effect size and variance [23,34,56]. Changes in Box and Blocks performance are described at individual and group levels with Hedge's G effect size and 95% CI.

For *scientific feasibility* related to Activities and Participation measures, the effect size (Hedge's g) and 95% CI are reported for changes in functional bimanual performance (AHA, Assisting Hand Assessment). Effect size for non-parametric data (COPM and SEAS) are reported using matched-pairs rank-biserial correlation (r) with 95% CI by bootstrapping [59]. These data analyses were completed in R (v. 3.7) employing packages: SingleCaseES [60] and RcmdrPlugin [23].

## Results

### Participant and recruitment characteristics

Fig 1 shows the CONSORT recruitment flow chart. Participant enrolment started September 2018 and completed January 2019. The target number of participants was reached. Table 1

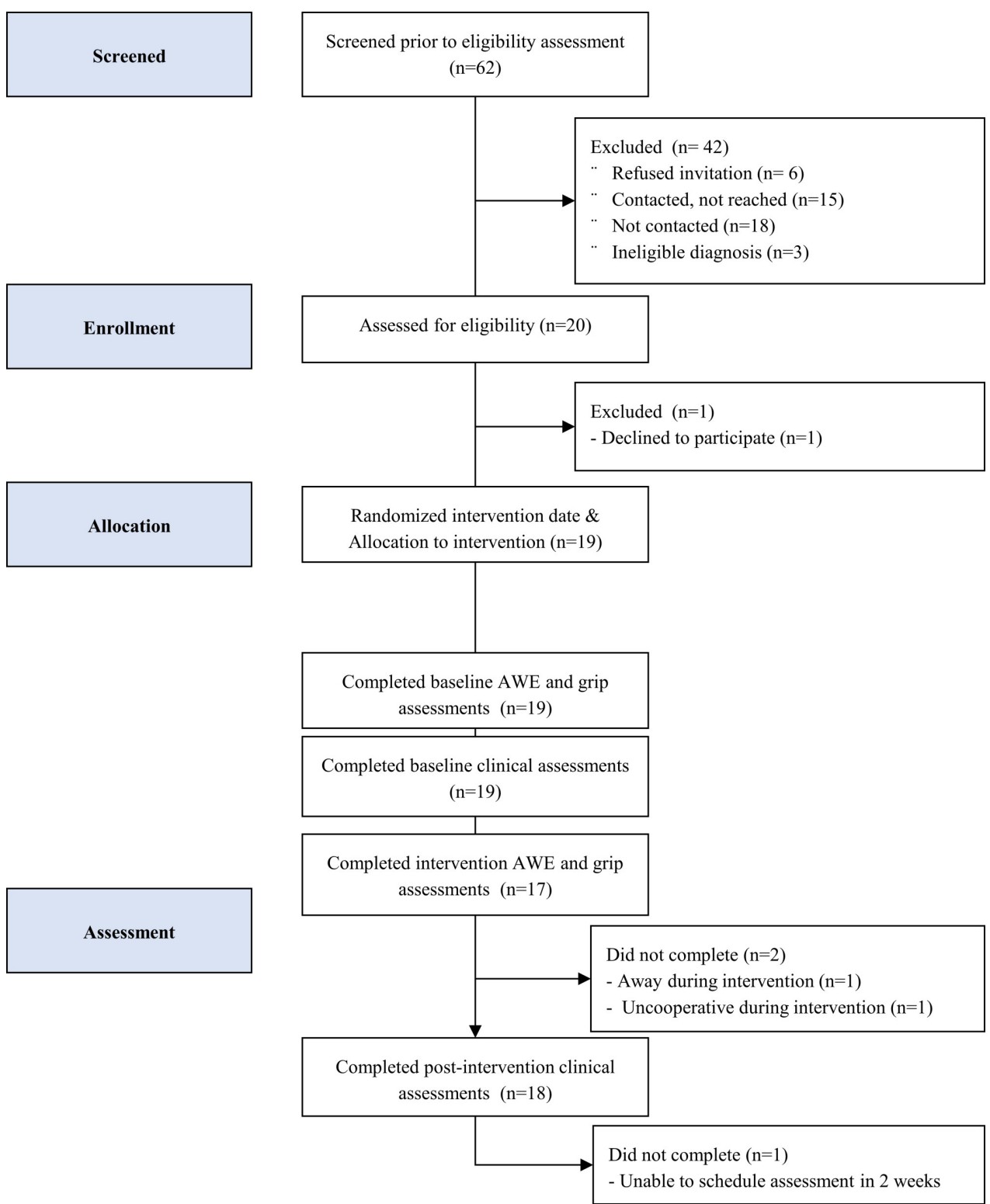

**Fig 1. CONSORT recruitment flow chart.** Standardized recruitment flowchart depicting the number of screened, enrolled, allocated and assess participants. Clinical assessment completed before and after intervention with occupational therapist: Box and Blocks, Assisting Hand Assessment and COPM Goals. As in single-case design, all participants allocated to same intervention.

**Table 1. Baseline demographic and clinical characteristics.**

| ID | Age | Sex | MACS | Affected Side | CP Type | Notes |
|---|---|---|---|---|---|---|
| A | 14.0 | M | I | L | SH | Control gesture: pinch |
| B | 13.0 | F | I | R | SH | Physiotherapy once weekly (lower limb) |
| C | 9.8 | M | I | R | SH | Learning disability |
| D | 9.8 | F | I | R | SH | - |
| E | 16.2 | F | II | R | SH | - |
| F | 10.5 | F | I | R | SH | Physiotherapy once weekly (upper/lower limb), control gesture: pinch |
| G | 10.5 | F | I | R | SH | - |
| H | 9.9 | F | II | R | MT | - |
| I | 13.8 | M | I | L | SH | - |
| J | 9.7 | M | II | R | MT | Learning disability |
| K | 10.4 | F | I | L | SH | Features of ADHD, intellectual disability, speech and language delays |
| L | 10.9 | M | II | R | MT | ADHD, seizure disorder, learning disability |
| M | 8.4 | M | I | L | MD | ADHD |
| N | 12.9 | M | II | R | MT | ASD, epilepsy |
| O | 8.4 | F | I | R | MT | - |
| P | 14.5 | M | II | R | MT | ADHD |
| Q | 17.4 | F | I | R | MT | - |
| R | 11.8 | F | II | R | SH | - |
| S | 9.8 | M | I | L | SH | Learning disability |

Notes describe secondary diagnoses and comments reported by therapists. All other control gestures were wrist extension- open fingers, wrist extension- closed fingers. Abbreviations: Mild Dystonia-MD, Spastic Hemiplegia-SH, Mixed tone-MT, attention deficit hyperactivity disorder-ADHD, Manual Abilities Classification System-MACS, Right-R, Left-L.

shows baseline demographic and clinical characteristics. System usage varied across individuals but averaged 4±1 days/week (8–24 days total), 17±9 minutes/day (37–333 minutes total), and 163±59 gesture repetitions/day (997–5698 total). No adverse events were reported. Two participants reported mild forearm muscle soreness during a weekly check-in. The soreness lasted for one day and resolved naturally without intervention.

## Process feasibility success criteria

Table 2 summarizes the *a priori* success criteria evaluation. As most (i.e. recruitment rate, completion), but not all criteria (i.e. frequency of technical issues) were met, the recommendation is 'feasible with minor modifications'.

## Scientific feasibility

Table 3 shows individual system utilization (dose) and scores for the six person-centred outcome measures.

**Table 2. Feasibility success criteria evaluation.**

| Criteria | Percent achieved | Evaluation description | Criteria met |
|---|---|---|---|
| ≥10% response rate | 31% | 19/62 of eligible participants were recruited | Yes |
| ≥80% complete study | 84% | 3/19 participants completed all assessments | Yes |
| ≥66% of the self-identified practice goals met | 74% | 14/19 participants met goal criteria* | Partial |
| 0 practice restrictions from technical issues | 17% | 6/36 reported technical issues not resolved immediately and restricting practice | No |

* Partial completion as some but not all participants (74%) reached ≥66% of the self-identified practice goals.

**Table 3. Individual system usage and outcomes.**

| | | Dose | | | Response | | | | | | | |
|---|---|---|---|---|---|---|---|---|---|---|---|---|
| | | Session does (median, IQR) | | | Body Function | | | Activities and Participation | | | | |
| | | Reps. | Minutes Active | Minutes in System | AWEO (°) | Grip (/1) | B&B (blocks) | AHA (logit) | COPM-P (/10) | COPM-S (/10) | SEAS (+3 - -3) | |
| ID | Days | | | | S \| F | S \| F | S \| F | S \| F | S \| F | S \| F | S \| F | |
| A | 12 | 114 (161) | 7.1 (11) | 10.5 (12) | 36 \| 50 | 0.5 \| 0.48 | 30 \| 39 | 62 \| 57 | 3 \| 2 | 2 \| 3 | 3 \| 2 | |
| B | 21 | 238 (322) | 15.6 (17.7) | 29.7 (21.6) | 26 \| 45 | 0.38 \| 0.53 | 16 \| 24 | 55 \| 54 | 4 \| 7 | 4 \| 5 | 2 \| 2 | |
| C | 18 | 74 (75) | 4.3 (5) | 12.5 (12) | 36 \| 50 | 0.5 \| 0.46 | 16 \| 17 | 43 \| 43 | 1 \| 3 | 1 \| 5 | 2 \| 2 | |
| D | 18 | 90 (149) | 4.1 (6.6) | 16.5 (19.3) | 22 \| 35 | 0.47 \| 0.58 | 16 \| 23 | 55 \| 55 | 4 \| 4 | 4 \| 2 | 0 \| 0 | |
| E | 24 | 186 (169) | 6.1 (5.8) | 15.3 (17.7) | -23 \| -5 | 0.19 \| 0.19 | 11 \| 8 | 43 \| 46 | 4 \| 3 | 2 \| 8 | 1 \| 2 | |
| F | 21 | 68 (119) | 7.2 (14.7) | 25.2 (24.3) | 9 \| 25 | 0.24 \| 0.45 | 26 \| 29 | 54 \| 55 | 7 \| 8 | 8 \| 7 | 2 \| 2 | |
| G | 20 | 103 (123) | 7.6 (15.9) | 23.8 (15.5) | -8 \| 20 | 0.51 \| 0.64 | 19 \| 18 | 64 \| 70 | 2 \| 5 | 3 \| 10 | -2.5 \| -2 | |
| H | 14 | 186 (182) | 6.9 (7.1) | 12.7 (17.7) | -14 \| 20 | 0.42 \| 0.47 | 5 \| 3 | 52 \| 54 | 3 \| 7 | 3 \| 10 | 3 \| 3 | |
| I | 11 | 65 (50) | 3 (2.5) | 7.2 (8.7) | -5 \| - | 0.20 \| - | 17 \| - | 48 \| - | 3 \| - | 3 \| - | 3 \| 2 | |
| J | 14 | 174 (192) | 6.2 (4.6) | 9.8 (7.4) | -20 \| 0 | 0.44 \| 0.44 | 16 \| 16 | 57 \| 48 | 3 \| 5 | 0 \| 0 | 2 \| 1 | |
| K | 21 | 143 (82) | 8.3 (9.8) | 13.6 (14.6) | 30 \| 35 | 0.61 \| 0.76 | 22 \| 25 | 55 \| 63 | 5 \| 8 | 3 \| 8 | 3 \| 3 | |
| L | 12 | 210 (212) | 6.4 (6.7) | 23.5 (13.9) | 11 \| 30 | 0.51 \| 0.57 | 22 \| 26 | 48 \| 50 | 5 \| 6 | 5 \| 6 | 2 \| 2 | |
| M | 16 | 117 (192) | 5.2 (4.2) | 9.3 (11.3) | 61 \| 55 | 0.73 \| 0.78 | 31 \| 31 | 76 \| 77 | 5 \| 10 | 5 \| 10 | 2.5 \| 2 | |
| N | 14 | 50 (82) | 4.9 (6) | 10 (8.1) | - \| - | - \| - | 10 \| 13 | 32 \| 40 | 5 \| 6 | 8 \| 6 | 3 \| 2 | |
| O | 12 | 196 (258) | 6.3 (7.5) | 10.8 (22.8) | 35 \| 45 | 0.53 \| 0.58 | 32 \| 41 | 77 \| 79 | 4 \| 5 | 5 \| 5 | 3 \| 3 | |
| P | 14 | 112 (90) | 5.1 (3.6) | 7 (5.3) | 13 \| 10 | 0.39 \| 0.49 | 13 \| 15 | 50 \| 54 | 4 \| 4 | 4 \| 2 | 1 \| 2 | |
| Q | 14 | 79 (39) | 4.5 (2.9) | 10.2 (10.8) | 41 \| 40 | 0.42 \| 0.55 | 20 \| 17 | 59 \| 57 | 2 \| 4 | 1 \| - | 2 \| 2 | |
| R | 17 | 83 (141) | 3.9 (9.2) | 11.7 (10.7) | -16 \| 5 | 0.43 \| 0.32 | 24 \| 33 | 48 \| 50 | 4 \| 4 | 4 \| 2 | 2 \| 1 | |
| S | 8 | 205 (128) | 8.2 (6.4) | 13.3 (11.9) | 23 \| 25 | 0.42 \| 0.48 | 20 \| 17 | 62 \| 62 | 5 \| 3 | 5 \| 1 | 3 \| 3 | |

Participants system usage in total days played (Days), average daily repetitions (Reps.), average time spent actively playing in the system (Minutes Active), average time using the system (Minutes in System). Body Function and Activities and Participation measures are presented. Starting (S) scores are: the median values at baseline for AWEO (Active wrist extension–open fingers, positive values indicate extension above neutral) and Grip (non-dominant grip strength relative to dominant), therapist assessed baseline values for B&B (Box and Blocks Test) and AHA (Assisting Hand Assessment- Logit score/100), COPM-P (primary goal's performance score), COPM-S (primary goal's satisfaction score), and SEAS (overall score on 7-point Likert scale) assessed at first week of play. Finishing (F) score are median score during final week of the intervention for AWEO and Grip, therapist assessment within 2-weeks of the end of intervention for B&B, AHA and COPM, and SEAS after the final day of practice. Insufficient/ not collected data denoted by -.

**Body function.** Active wrist extension (AWEO) increased 12±12˚. There was a moderate to large effect for AWEO (SMD = 1.82, 95%CI = 0.85–2.78). Grip strength also increased (17 ±18%) and there was a small to moderate effect (SMD = 0.63, 95%CI = 0.65–1.91). Fig 2 shows the number of participants with small, moderate and large effects through Slope and Level change, and PAND analysis. A positive increase in at least one Body Function measure was seen in 14/19 participants (see S7 File, for slope and level changes for each participant).

Change score for non-dominant Box & Blocks performance showed a moderate effect (Hedge's g = 0.58, 95%CI = -0.11–1.27). Box & Blocks scores were between 5–32 at pre-test and 3–41 at post with a median change of +2.5 blocks (Table 3).

**Activities and participation.** Fig 3 summarizes pre-post changes, ordered by practice time (minutes in system). Post-hoc analyses showed small to no relationship between practice

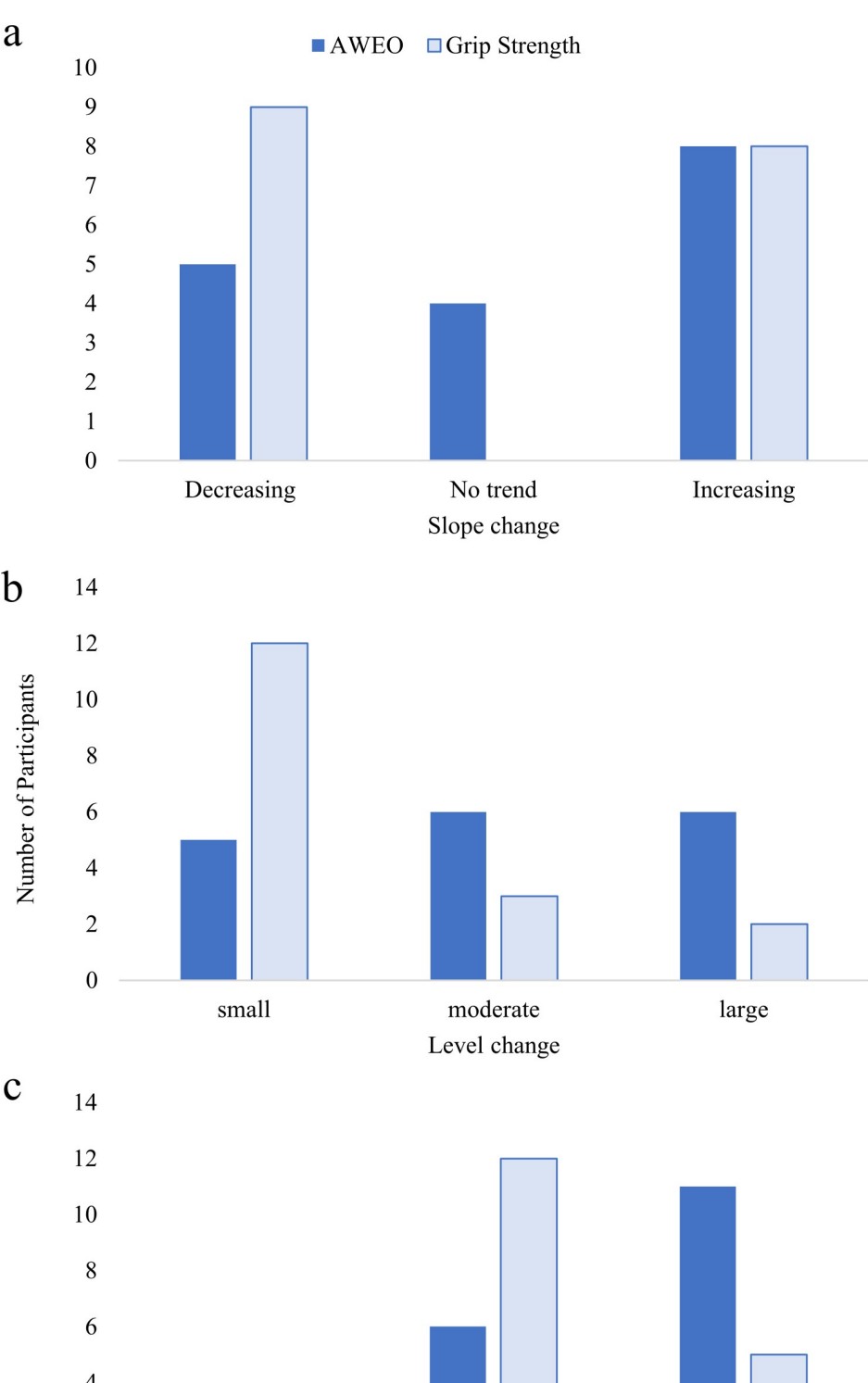

**Fig 2. Visual analyses summary for slope, level and non-overlapping data.** Number of participants showing changes between baseline and intervention phases (total N = 17). Active wrist extension–open fingers (AWEO, dark) and grip strength (light). (a) Slope changes, increasing indicates intervention phase slope is greater than baseline slope. (b) Level changes determined by split-middle method, small—Intervention phase < 5˚ (AWEO) or <5% (Grip Strength) from Baseline, moderate—Intervention phase 5–15˚ (AWEO) or 5–15% (Grip Strength) from Baseline, and large—Intervention phase >15˚ (AWEO) or >15% (Grip Strength) from Baseline. (c) Percent of all non-overlapping data, 50–80% of all non-overlapping data between phases indicates moderate separation between baseline and intervention and >80% of all non-overlapping data between phases indicates large separation between baseline and intervention.

time and functional change scores (B&B, r = 0.19; AHA, r = 0.20; COPM, r = 0.30; SEAS, r = 0.08). Four weeks of the intervention showed a small effect in AHA score (Hedge's g = 0.29, 95%CI = -0.39–0.97). AHA scores ranged between 32–77 at pre-test and 40–79 at post with a median change of +1.5 logit units. There was a moderate effect for COPM Performance scores (r = 0.60, 95%CI = 0.13–0.82). Median COPM change was +1 post intervention ranging from -2–5. See S1 Table for COPM Goals. The SEAS questionnaire showed participants felt positively about the activity (median = +2, IQR = 1.25) at the beginning and end of

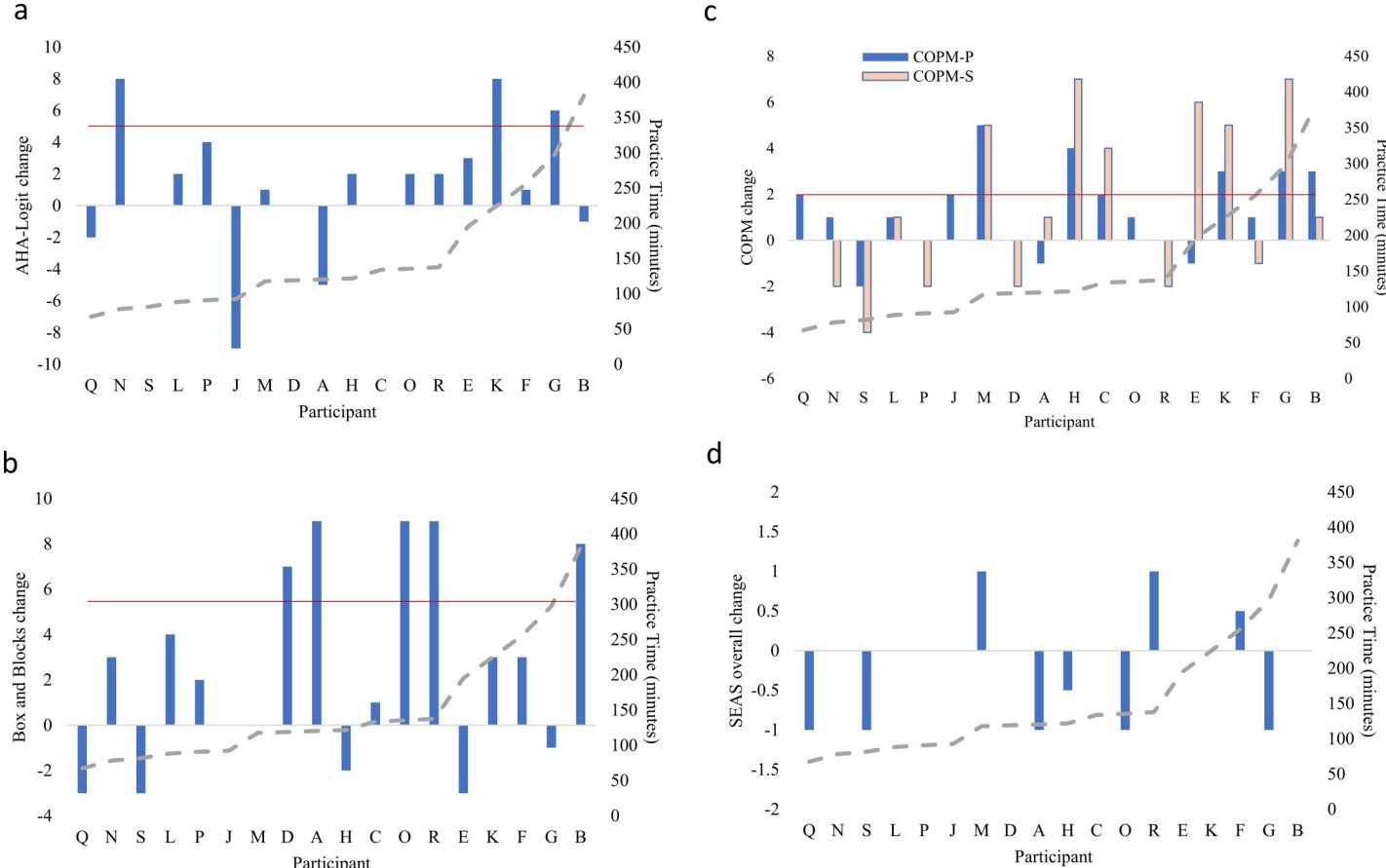

**Fig 3. Pre- post-intervention change scores by participant.** Differences in score before and after intervention for each measure, positive values indicate higher post-intervention score. (a) Assisting Hand Assessment (AHA) logit units, b) Box and Blocks (Blocks / minute), c) Canadian Occupational Performance Measure (COPM) P-performance, S-satisfaction (1–10 scale), and d) Self-Reported Experiences of Activity Settings (SEAS) overall score (+3- -3, 7-point Likert scale). Solid horizontal line indicates clinically meaningful change. Red, horizontal lines indicate clinically meaningful difference where available. Participants arranged left to right by total practice time (minutes) as indicated by dotted line, showing little correlation (r = 0.8–0.3) between practice time and change score for each measure.

the intervention. SEAS score did not change (Median change = 0, range = -1–1, IQR = 0.75, small effect r = 0.24, 95%CI = —.25–0.61). See S2 Table for SEAS subscale scores.

## Discussion

This study assesses the feasibility of novel intervention combining Solution Focused Coaching strategies with biofeedback-enhanced movement-controlled gaming. Most but not all *a priori* success criteria were met, as such the intervention approach is *feasible with modifications*, most notably in the refinement of the technology to mitigate technical issues. Clinical outcomes of the intervention were promising with moderate effects in Body Function measures and small-moderate effects in Activities and Participation measures.

### Recruitment and adherence

This study shows comparable recruitment and retention rates to similar interventions. When reported, previous home-based ICP interventions have shown 37–46% recruitment of eligible participants and 66–90% retention through the intervention [38,61–63]. Adherence was relatively high compared to other studies which have shown participants complete 53–78% of practice goals during intervention periods 4–20 weeks long [38,61–63]. Few studies transparently report relevant metrics of the practice (timing, duration intensity) which makes it difficult to quantify and compare the therapy dose between studies. Standardized reporting in this area would facilitate meta-analyses needed to strengthen the evidence for home-based gaming interventions. At the observed effect size, 0.29, in bimanual performance (AHA change score), 127 participants would be required for the definitive RCT assuming alpha significance set to 0.05 and a desired power = 0.9. AHA change score was chosen to estimate the sample size instead of active wrist extension capacity since it measures functional bimanual performance and has a more direct relationship with ability to perform activities of daily living.

### Implementation

It is important to note that this is the first investigation of the novel home-based gaming intervention and accordingly focused on *process* and *scientific feasibility*. Specifically, the study focuses on determining the extent to which the intervention could be used at home and estimating the effect size and variance it might have. Towards conducting a complete RCT, more studies and development would be needed to ascertain real-world feasibility, addressing resources and management. This includes a proper health economics analysis and evaluation of the logistic organization with respect to an institute's existing therapeutic practices. It should also be noted that participants in Canada and France were off treatment blocks but visited their care provider for regularly scheduled check-ups, either annually or quarterly. Two participants, B and F, took additional weekly therapy sessions outside of the rehabilitation centre and one participant, J, attended the school at the rehabilitation centre and would see the occupational therapist as needed.

### Motivation

We expect that the coaching and biofeedback strategies helped maintain participants' engagement in the intervention. Biofeedback was linked to a variety of short- and long-term goals in the game. Participants' motivations were linked to game biofeedback. Participants regularly chose to review their progress and adapted their movements in response to game feedback (S1 File). The Solution Focused Coaching strategy helped to maintain cognitive engagement in the intervention, reiterating how the game addressed functional goals. The positive, self-directed

rhetoric was more effective with participants who believed that the game addressed their functional goals. Five of 19 participants did not reach their practice goals. These participants showed a novelty effect, with little interest after 1–2 weeks with the new game. These participants commented that they felt the activity did not align with their functional goals (*e.g. Spread thumb easier to use joystick when playing video games*) indicating a lack of cognitive engagement. Participants who were less convinced of the relevance of the task found it more difficult to participate in the coaching conversations and verbalize how their success in the game could translate to daily activities. In such cases, we explored alternative motivation strategies (*e.g.* parent-identified rewards for adhering to practice goals, or leader boards playing to a competitive nature). The SFC strategy required flexibility in the conversation and training structure. We would recommend it as a tool to engage participants but would not rely solely on SFC. Overall our learning in this study defends the importance of ensuring that home-based ICP therapy activities align with individual motivations and goals to support cognitive, affective and behavioural engagement in the intervention [14].

## Body function

Active wrist extension is moderately to highly related to manual abilities [43,64]. In the current study, 12 of 17 participants increased active wrist extension by at least 5 degrees. These findings are consistent with other home-based supplemental therapy activities. Comparatively, [28] found 6±3 degrees improvement in wrist extension across 30 young children with CP after 24 weeks at 3 * 60 minutes/week and [27] saw 18±12 degrees change with four participants after 30 minutes * 5 sessions of EMG-based neurotherapy. Note, the wide range in practice time and different nature of the interventions may contribute to the inconsistency between studies.

Wrist extension capacity is also directly related to grip strength [65]. Accordingly, we saw 18% average improvement in non-dominant grip strength relative to the dominant side across participants. However, normative data for children's grip strength, as compared to the dominant side using the modified sphygmomanometer are not available. For comparison, [66] reported an average 15% grip strength improvement after 12-weeks hand function training in 15 children with CP. [67] observed a median 25% change from 5 participants pre-test values after 8-weeks of single joint resistance training combined with Botulinum toxin A injections.

Post-hoc analyses showed differences in Body Function effect based on CP severity. Participants with more severe involvement (MACS level II) had greater gains in active wrist extension (8±10°) and smaller gains in grip strength (-8±7%) compared to participants at MACS level I. Statistical confirmation is not advised based on sample size and variance. More severely affected participants, those at MACS II had below neutral maximum wrist extension. Therefore, there was more opportunity for amplitude improvement, but without being in an extended posture, it is difficult to optimize grip strength.

[68] established through path analysis from records of 136 children with CP that grip strength indirectly contributes to manual ability (Abilhand-Kids) [69] via its influence on gross manual dexterity (B&B) [68]. Consequently, active wrist extension and grip strength may indicate changes in manual ability relevant to daily activities. Despite this, we recognize the increased focus and relevance towards Activity and Participation measures [70]. Accordingly, the Activity and Participation measures were included in this early-stage feasibility study.

## Activities and participation

There were small effects in Activities and Participation related measures (AHA, COPM). A minority of participants met or exceeded clinically meaningful thresholds (3/19 for AHA and

8/19 for COPM), while the majority showed non-clinically significant positive changes. The small effect is most likely due to low dosage and the nature of the activity. Further considering this relatively small dose across all participants, it is not surprising to see small correlations between practice time and functional change scores. The biofeedback video game practices specific functional movements but it is not an activity-based intervention (e.g. Constraint Induced Movement Therapy) [41]. Since practical manual ability is not the simple summation of skill and structure it would be unreasonable to expect gross transfer to daily tasks. However, considering the need for diverse and engaging rehabilitation strategies and the relative low risk of harm of this ICP intervention it may be a useful supplement to activity-based interventions. It may help accelerate Body Function changes (e.g. active wrist extension, grip strength) to facilitate manual ability improvements. This question could be addressed in future clinical trials.

The SEAS questionnaire showed that participants experience was consistent during the 4-weeks. This corresponds with observations during the weekly check-in conversations. Only in the five participants who experienced a novelty effect, as described above, did we observe a decrease in the sub-scale score, Psychological Engagement (S2 Table).

## Limitations

Due to resource limitations, a single game was built which may not have optimally appealed to the wide range in ages (8–18 years) and interests of the participants. General interest in video games or this game was not an inclusion criterion. Considering the impact personal motivation has on adherence and the vast differences in personal preferences, it is essential to match participants to activities that interest them. Greater choice and game variety, while challenging to implement in rehabilitation protocols, would help maintain novelty and interest in the activity. Collaboration with independent game developers can improve feasibility by offering content with relatively quick and flexible modification abilities, as was the case in the current study with the adapted commercial video game (Dashy Square) [71].

Methodologically, there was risk of bias in assessment scoring as clinicians and researchers were not blinded to the participant phase. Bias in goal setting is also possible as parents, clinicians and researchers were present when the participants set their practice and COPM goals. Further, COPM responses are subjective and can be influenced by mood and environment. For instance, one participant successfully completed her goal for the first time at the post-intervention assessment but scored the performance lower than at baseline. For these reasons we use multiple measures to capture Activities and Participation experiences. Next, level-change groupings (small, moderate, and large) for active wrist extension were based on a minimal detectible difference of five degrees [42]. However, similar level-change groupings in grip strength were undeterminable since minimal detectable differences for grip strength, as compared to the dominant hand, do not yet exist for young people with CP. Scores are reported as a percent of the dominant side since improved capacity for bimanual activities is a primary goal for many young people with CP. For context, raw score increases in the affected hand's grip strength averaged 17 mmHg and all but two participants saw an improvement of greater than 10 mmHg. A minimal detectable difference of seven mmHg and a within-subject Standard Error of Measurement of three mmHg has recently been reported in individuals with Parkinson's Disease using a similar modified sphygmomanometer test [72]. These visual analyses summaries are not provided as a definitive evaluation but to aid the reader in their interpretation of the effect size in this single-case design intervention [23]. Further, while AB designs are useful for evaluating feasibility, return-to-baseline, or withdrawal designs would improve the strength of evidence of treatment effects [23]. The SEAS questionnaire was a

practical tool to implement in the home to gauge self-reported experience. For a comprehensive evaluation, future work should consider qualitative interviews and content analysis [73]. The Solution Focused Coaching approach is designed to encourage collaborative development, led by the participant but does acknowledge the potential for external influence [16]. Here we kept fidelity of the SFC approach by referring to a checklist, but this could be improved by video review and completing a fidelity questionnaire [16].

Logistically, the protocol may benefit from increased clinician involvement. Occupational Therapists remarked that that they could have helped guide home-based training by observing participants playing at weekly visits or by video. Finally, this study used the AHA as an outcome measure focusing on the non-dominant hand's involvement in bimanual activities. Other measures of manual performance (i.e. Melbourne Assessment of Unilateral Upper Limb Function) [46] have been proposed and used more widely. Changing this metric could facilitate cross-study comparison.

## Supporting information

**S1 File. Biofeedback technology development.**
(PDF)

**S2 File. The Single-Case Reporting Guideline In BEhavioural interventions (SCRIBE) 2016 checklist.**
(PDF)

**S3 File. The Consolidated Standards of Reporting Trials (CONSORT) checklist.**
(PDF)

**S4 File. Procedural fidelity resource.**
(PDF)

**S5 File. Template for Intervention Description and Replication (TIDieR) checklist.**
(PDF)

**S6 File. Intervention protocol.**
(PDF)

**S7 File. Individual visual analysis.**
(PDF)

**S1 Table. COPM goals.**
(DOCX)

**S2 Table. SEAS subscale scores.**
(DOCX)

## Acknowledgments

We are greatly appreciative of the time and input of the participants and families who helped us build and test this system. The authors wish to acknowledge the occupational therapists and research administration staff who supported this project. Special thanks to the head of the DISI, and to the head of the UFR ST, of University of Evry for lending computers. And to the developer of the game, KasSanity Inc. The authors also wish to acknowledge the data processing and administrative assistance of the Research Manager.

## Author Contributions

**Conceptualization:** Alexander MacIntosh, Eric Desailly, Nicolas Vignais, Vincent Vigneron, Elaine Biddiss.

**Data curation:** Alexander MacIntosh, Elaine Biddiss.

**Formal analysis:** Alexander MacIntosh.

**Funding acquisition:** Alexander MacIntosh, Nicolas Vignais, Elaine Biddiss.

**Investigation:** Alexander MacIntosh.

**Methodology:** Alexander MacIntosh, Eric Desailly, Nicolas Vignais, Elaine Biddiss.

**Project administration:** Alexander MacIntosh, Eric Desailly, Nicolas Vignais, Vincent Vigneron, Elaine Biddiss.

**Resources:** Alexander MacIntosh, Eric Desailly, Nicolas Vignais, Vincent Vigneron, Elaine Biddiss.

**Software:** Alexander MacIntosh, Nicolas Vignais.

**Supervision:** Alexander MacIntosh, Eric Desailly, Nicolas Vignais, Vincent Vigneron, Elaine Biddiss.

**Validation:** Alexander MacIntosh.

**Visualization:** Alexander MacIntosh.

**Writing – original draft:** Alexander MacIntosh.

**Writing – review & editing:** Alexander MacIntosh, Eric Desailly, Nicolas Vignais, Vincent Vigneron, Elaine Biddiss.

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
