## [Decision Letter · Decision Letter 0]

3 Dec 2019

PONE-D-19-22575

A biofeedback-enhanced therapeutic exercise video game intervention for young people with cerebral palsy: a randomized single-case experimental design feasibility study.

PLOS ONE

Dear Alexander MacIntosh,

Thank you for submitting your manuscript to PLOS ONE. After careful consideration, we feel that it has merit but does not fully meet PLOS ONE’s publication criteria as it currently stands. Therefore, we invite you to submit a revised version of the manuscript that addresses the points raised during the review process.

the coherence between the study declared goal and the actual presentation of results has been questioned, as well as the way in which the results are presented and the significance evaluated- Please address these concerns properly. 

We would appreciate receiving your revised manuscript by January 15 2020. To enhance the reproducibility of your results, we recommend that if applicable you deposit your laboratory protocols in protocols.io, where a protocol can be assigned its own identifier (DOI) such that it can be cited independently in the future. For instructions see: http://journals.plos.org/plosone/s/submission-guidelines#loc-laboratory-protocols

We look forward to receiving your revised manuscript.

Kind regards,

Andrea Martinuzzi

Academic Editor

PLOS ONE

Journal Requirements:

2. Thank you for submitting your clinical trial to PLOS ONE and for providing the name of the registry and the registration number. The information in the registry entry suggests that your trial was registered after patient recruitment began. PLOS ONE strongly encourages authors to register all trials before recruiting the first participant in a study.

1) your reasons for your delay in registering this study (after enrolment of participants started);

2) confirmation that all related trials are registered by stating: “The authors confirm that all ongoing and related trials for this drug/intervention are registered”.

Please also ensure you report the date at which the ethics committee approved the study as well as the complete date range for patient recruitment and follow-up in the Methods section of your manuscript, and clarify why the protocol is dated 23 November 2018.

Reviewers' comments:

Reviewer's Responses to Questions

**Comments to the Author**

1. Is the manuscript technically sound, and do the data support the conclusions?

Reviewer #1: Yes

Reviewer #2: Partly

2. Has the statistical analysis been performed appropriately and rigorously? 

Reviewer #1: Yes

Reviewer #2: No

3. Have the authors made all data underlying the findings in their manuscript fully available?

Reviewer #1: Yes

Reviewer #2: Yes

4. Is the manuscript presented in an intelligible fashion and written in standard English?

Reviewer #1: Yes

Reviewer #2: No

5. Review Comments to the Author

Reviewer #1: The manuscript describes a technically sound piece of scientific research.

The statistic analysis has been conducted rigorously

The data support the conclusions and I totally agree with the resource limitation well described at paragraph 430

Reviewer #2: General comment

The authors present the results of a study to assess the feasibility of an at-home video game intervention. However, the authors seem to mix a bit the purposes of this study and the paper is difficult to follow. The main aim is to determine the feasibility (adherence, complications) but then the authors also assess the motor functions of the partcipants before and after the intervention. Then they used unusual methods for presenting the results. The article would become more readable if the authors only present the results of the feasibility part and shorten the presentation of the pre-post results since it’s not the aim of this particular study. A table with the results of the different tests before and after the intervention with results of the statistics would be more suited and easy to interpret, the other analysis can be moved to supplementary materials.

Specific comments

Introduction

Line 45: The authors should precise the inconsistency in the existing studies and what is currently missing to strengthen their study.

Line 68: patient instead of client

Methods

Intervention: Was the total amount of training (defined during the self-defined practice) significantly different between the participants? The total duration of training should be presented in the results.

Results

Table 1: only 2 patients have physiotherapy sessions? It seems low compared to the recommendation. Should be discussed.

Table 2: 19 patients but for the technical issues 6/39? Please clarify

Figure 2 – 3: Forest plots are a very unusual way of presenting individuals' data making the interpretation confusing! How did you compute CI for the SMD based only on 1 individual data? What’s the point of using a random effect model since it’s a controlled study? Participants should all have the same weight. A simple t-test (or Wilcoxon if the data are not normaly distributed, which is not precised) is more adapted (eventually with line plots if you want to present individual’s results).

Figure 4: again visual analyses, the authors state that there did not compute the significance but still present a lot a statistics, this is a bit confusing.

Was there any difference between the experiments in France and in Canada?

Discussion

Body function: Against it is a bit confusing to read “In the current study, all but two participants increased active wrist extension” since no proper statistics were done.

“Accordingly, we saw 18% average improvement with +5% gains in 12/19 participants.” Idem, what about the 7 other participants?

Since it is a feasibility study the author should discuss the feasibility of implementing such kind of interventions in “real life” conditions. Do they think it is possible to have the pre-session and the different control sessions organized in practice (and who is going to pay for that). This is particularly important since apparently only 2 of the 19 participants have physiotherapy sessions (so basically no intervention).

6. PLOS authors have the option to publish the peer review history of their article (what does this mean?). If published, this will include your full peer review and any attached files.

Reviewer #1: No

Reviewer #2: Yes: Bruno Bonnechère

---

## [Author Response · Author response to Decision Letter 0]

3 Jan 2020

The following response can also be found in the 'Response to Reviewers.docx'

 Editor’s comments 

1 The coherence between the study declared goal and the actual presentation of results has been questioned, as well as the way in which the results are presented, and the significance evaluated- Please address these concerns properly. 

Thank you for your comments. In our responses to reviewer 2 you will see that their concerns regarding the presentation of the results and the evaluation of significance have been addressed. 

In summary, regarding the main aim of this study, we assess feasibility according to recommendations from Thabane et al 2010, and the CONSORT extension for reporting feasibility studies (Eldridge et al 2016). This framework highlights that the aim of feasibility testing can be towards assessing any of four classifications: process, resources, management and scientific. “Scientific” here refers to treatment safety and estimation of treatment effect and its variance. Motor function assessments were completed for this reason. 

It is also worth noting that statistical analyses were intentionally not conducted. In fact, the feasibility reporting guide we follow specifically discourages emphasizing statistical significance in studies of this type mainly due to low powered samples. We therefore follow the Single-Case Reporting Guideline In BEhavioural Interventions (SCRIBE), which recommends the presentation of a mixed ‘visual and statistical’ approach (Manolov et al 2014). The methods are considered complementary rather than mutually exclusive. We acknowledge that there is no universal gold-standard for single-case design studies, and it is for this reason we provide a mixed analysis approach.

Separating primary and secondary aims seems to have caused this confusion. To clarify, we have explicitly stated which aspects of feasibility criteria we address and have: 

- Added context around feasibility to the introduction

- Re-organized the methods to show that the current investigation focuses on process and scientific feasibility.

- Added a summary table of results as Reviewer 2 requested

You can find the specific changes made to the manuscript in the response to Reviewer 2, general comments # 1 and 3.

2 Please ensure that your manuscript meets PLOS ONE's style requirements, including those for file naming. The PLOS ONE style templates can be found at

We have reviewed the style guidelines and confirm they meet the requirements.

3 Thank you for submitting your clinical trial to PLOS ONE and for providing the name of the registry and the registration number. The information in the registry entry suggests that your trial was registered after patient recruitment began. PLOS ONE strongly encourages authors to register all trials before recruiting the first participant in a study. 

Thank you for reviewing this. We apologize for the confusion here. We think this may be because the ‘Last Update’ date was posted June 18, 2019. This update was related to an annual renewal of the study protocol on the ClinicalTrials.gov registry.

The trial, registered at: https://clinicaltrials.gov/ct2/show/NCT03677193 indicates that this study was first posted September 19, 2018. Further, the initial release date according to the ClinicalTrials.gov Protocol Registration and Results System was dated July 31, 2018. Note, this is before the start of the study enrollment. 

Please see the screenshots at the bottom of this document to confirm (Appendix A). If for some reason, the study still appears to be registered after recruitment began, we will indeed contact the registry directly to correct the error. Thank you for bringing this to our attention. The trial registration number may have also been incorrect. We have verified that the proper registration number is in the manuscript.

4 As per the journal’s editorial policy, please include in the Methods section of your paper:

1) your reasons for your delay in registering this study (after enrolment of participants started);

2) confirmation that all related trials are registered by stating: “The authors confirm that all ongoing and related trials for this drug/intervention are registered”. 

Regarding item 1) as described in response #3, above, the study was registered July 31, 2018 on ClinicalTrials.gov. The first participant was enrolled on September 29th, 2019.

Regarding item 2) the required statement was added to the section: Methods, Design.

5 Please also ensure you report the date at which the ethics committee approved the study as well as the complete date range for patient recruitment and follow-up in the Methods section of your manuscript, and clarify why the protocol is dated 23 November 2018. Methods have been adjusted accordingly and now read as follows: 

“Ethical approval was obtained by Holland Bloorview’s Research Ethics Board (approved July 27, 2018, amended November 23, 2018, to include participants with mixed tone and mild dystonia) and the French national Comité de protection des personnes (CPP) (approved July 6, 2018).” – Section: Methods, Design

“Recruitment in Canada took place from September-November 2018. … Recruitment in France took place from November 2018 - January 2019.”- Section: Methods, Participants, Recruitment

Reviewer #1 comments: 

1 The manuscript describes a technically sound piece of scientific research.

The statistic analysis has been conducted rigorously

The data support the conclusions and I totally agree with the resource limitation well described at paragraph 430 

Thank you for your kind remarks.

Reviewer #2 

General comments 

1 The authors present the results of a study to assess the feasibility of an at-home video game intervention. However, the authors seem to mix a bit the purposes of this study and the paper is difficult to follow. The main aim is to determine the feasibility (adherence, complications) but then the authors also assess the motor functions of the participants before and after the intervention. 

We would first like to thank Reviewer 2 for their detailed comments. 

Regarding the main aim of this study, we assess feasibility according to recommendations from Thabane et al 2010, and the CONSORT extension for reporting feasibility studies (Eldridge et al 2016). This framework highlights that the aim of feasibility testing can relate to any of four broad classifications: process, resources, management and scientific. Scientific here refers to treatment safety and estimation of treatment effect and its variance. Motor function assessments were completed for this reason. We decided to separate the scientific, effect size and variance, estimate to the secondary aim as we used six outcome measures (since no single measure addresses all aspects of capacity and participation). However, we apologize, as this seems to have caused confusion. To clarify, all the reported outcomes, including those related to motor function are made towards assessing feasibility of the intervention. 

It is also worth noting that for this reason, statistical analyses were intentionally not conducted. In fact, the feasibility reporting guide we follow discourages emphasizing statistical significance in studies of this type (Thabane et al 2010, Eldridge et al 2016). The goal here is to inform future randomized controlled trial (RCT) research and not verify the significance of the response, as the study is underpowered to do so.

We have made changes in the main text to clarify our use of the feasibility framework: 

“This paper addresses intervention feasibility as articulated by Thabane et al (2010) [18]. This framework highlights that the aim of feasibility testing can be related to one or more of the following four classifications: process, resources, management and scientific. The objective here is to support development towards future randomized controlled trials and not to define statistical or clinical effectiveness of the intervention [18]. In this study, we concentrate on two of the four feasibility classifications: 

First, we assess process feasibility of the biofeedback-enhanced therapy video game intervention protocol for young people with CP. The objective here is to determine the ability to enroll participants, enable home-based practice, and retain their activity during a 1-month intervention. To this purpose, a priori success criteria were established for the recruitment and response rates, adherence, and frequency of technical difficulties impeding home practice [18]. 

Second, we assess the scientific feasibility of the intervention by estimating the effect size and variance for six participant-centred outcome measures for the hand and wrist. The measures are aligned to the Body Functions and Activities and Participation chapters of the ICF (international classification of functioning disability and health) [19,20].” – Section: Introduction, Aim

2 Then they used unusual methods for presenting the results. The article would become more readable if the authors only present the results of the feasibility part and shorten the presentation of the pre-post results since it’s not the aim of this particular study. 

As indicated in the comment above, we assess feasibility according to recommendations from Thabane et al 2010, and the CONSORT extension for reporting feasibility studies (Eldridge et al 2016). This framework highlights that the aim of feasibility testing can relate to any of four broad classifications: process, resources, management and scientific. Scientific here refers to treatment safety and estimation of treatment effect and its variance. Motor function assessments were completed for this reason. All the reported outcomes, including those related to motor function (the pre-post results) are made towards assessing feasibility of the intervention.

To clarify this, we have: 

- Added context around feasibility in the aim (please see the quoted text in the comment above). 

- Re-organized the methods to not distinguish the work in terms of primary and secondary aims, but in reference to process feasibility and scientific feasibility. You will see this change throughout the aim, methods and results of the manuscript. 

3 A table with the results of the different tests before and after the intervention with results of the statistics would be more suited and easy to interpret, the other analysis can be moved to supplementary materials. 

For clarity, we have provided a table of the results of the different tests. Please see Table 3 under the section: Results, Scientific Feasibility. 

Please note, as we followed the CONSORT extension for reporting feasibility studies (Eldridge et al 2016) and the Single-Case Reporting Guideline In BEhavioural Interventions (SCRIBE), we have intentionally not reported significance tests. We do however report the combined statistical and visual analysis along with estimates of effect size and their variance in the form of 95% confidence intervals where appropriate. 

We hope the reviewer can understand the choice to use this methodology, and not to focus on statistical significance. The goal here is to inform future RCT research and not verify the significance of the response, as the study is underpowered to do so. To this point we have added the following statement: 

“At the observed effect size, 0.29, in bimanual performance (AHA change score), 127 participants would be required for the definitive RCT assuming alpha significance set to 0.05 and a desired power = 0.9. AHA change score was chosen to estimate the sample size instead of active wrist extension capacity since it measures functional bimanual performance and has more direct relationship with ability to perform activities of daily living.” – Section: Discussion, Recruitment and adherence

Specific comments 

Introduction 

1 Line 45: The authors should precise the inconsistency in the existing studies and what is currently missing to strengthen their study. 

The statement has been changed to: 

“These types of studies have shown moderate evidence towards improving balance and overall motor skill but weak evidence towards improving upper extremity skills, joint control, gait and strength [9]” - Introduction.

2 Line 68: patient instead of client

 We have removed the use of the terms: patient and client throughout the manuscript. Our institution’s convention is to use client instead of patient, however we realize this is not true everywhere. To be as inclusive as possible we have used the term participant, person or individual where 

Methods 

3 Intervention: Was the total amount of training (defined during the self-defined practice) significantly different between the participants? The total duration of training should be presented in the results. 

Individual practice did vary across participants. While the average amount of training was 16±4 days over the 1-month intervention, 7/19 participants were outside 1 standard deviation from this mean. Similarly, while participants practiced an average of 17±9 minutes/day, 10/19 participants were outside 1 standard deviation from this mean. 

Individual total amount of training can now be seen in Table 3 under section: Results, Scientific feasibility. We have also added the following statement: 

“System usage varied across individuals but averaged 4±1 days/week (8-24 days total), 17±9 minutes/day (37-333 minutes total), and 163±59 gesture repetitions/day (997-5698 total).” - Section: Results, Participant and recruitment characteristics

Results 

4 Table 1: only 2 patients have physiotherapy sessions? It seems low compared to the recommendation. Should be discussed. 

We have added the following to the discussion: 

“Participants in Canada and France were off treatment blocks but visited their care provider for regularly scheduled check-ups, either annually or quarterly. Two participants, B and F, took additional weekly therapy sessions outside of the rehabilitation centre and one participant, J, attended the school at the rehabilitation centre and would see the occupational therapist as needed.” - Section: Discussion, Implementation 

Both Canada and France have publicly funded healthcare but services for individuals in our recruitment demographic (older age group and mild to moderate impairment) are not as frequently utilized. As such, it is not uncommon to see older children and teenagers with CP only occasionally be on active blocks of manual therapy training. 

5 Table 2: 19 patients but for the technical issues 6/39? Please clarify 

Technical issues could be reported multiple times during the 1-month intervention. For instance, a participant might report a technical issue once in the first week and twice more in the third week. As such, it is possible to have more technical issues than participants enrolled.

To clarify this metric, we have added the following text to Methods, Outcomes #4:

“Participants may report multiple issues during the 1-month intervention.”

6 Figure 2 – 3: Forest plots are a very unusual way of presenting individuals' data making the interpretation confusing! How did you compute CI for the SMD based only on 1 individual data? What’s the point of using a random effect model since it’s a controlled study? Participants should all have the same weight. A simple t-test (or Wilcoxon if the data are not normally distributed, which is not precised) is more adapted (eventually with line plots if you want to present individual’s results). 

Thank you for pointing this out. As described in the response to this reviewer’s general comments numbers 1 and 3, we are not presenting statistical tests such as t-tests or Wilcoxon to conform with the CONSORT feasibility testing and SCRIBE Single-Case design reporting frameworks.

We do however acknowledge that this is an unusual presentation of individual results. Given the individual outcomes are now reported in Table 3, we have removed Fig 2 and Fig 3 and reported group level SMD and confidence intervals descriptively (Section: Methods, Scientific feasibility). 

7 Figure 4: again visual analyses, the authors state that there did not compute the significance but still present a lot a statistics, this is a bit confusing. 

As described in the response to this reviewer’s general comments numbers 1 and 3, significance tests comparing pre and post outcomes were not conducted in accordance with the CONSORT feasibility testing and SCRIBE Single-Case design reporting frameworks.

However, statistical analyses were conducted to establish effect sizes and variances as recommended by CONSORT feasibility testing and SCRIBE Single-Case design reporting. SCRIBE recommends the presentation of a mixed ‘visual and statistical’ approach (Manolov et al 2014). The methods are considered complementary rather than mutually exclusive. We acknowledge that there is no universal gold-standard for SCED design studies, and it is for this reason we provide a mixed analysis approach.

8 Was there any difference between the experiments in France and in Canada? 

No, the experiments in France and Canada were completed the same way. The same researcher trained the participants, completed all in-home assessments and measures of range of motion and grip strength. The only difference was in recruitment where:

“In Canada, eligible participants were also identified through the hospital’s centralized recruitment database, connect2research. In Canada the researcher telephoned potential participants after sending an invitation letter by mail. Then, the researcher screened interested participants and obtained written informed consent. In France, the developmental pediatricians invited eligible individuals to participate. They obtained written informed consent from interested participants.” 

This is currently expressed in the section: Methods, Participants, Recruitment. 

Discussion 

9 Body function: Against it is a bit confusing to read “In the current study, all but two participants increased active wrist extension” since no proper statistics were done. 

This statement has been further specified for clarity and now reads: 

“In the current study, 12 of 17 participants increased active wrist extension by at least 5 degrees.” – Section: Discussion, Body Function

Additionally, we have added the following to the methods: 

“...values are recorded to the nearest five degrees, the minimal detectable difference [42].” – Section: Methods, Outcomes

10 “Accordingly, we saw 18% average improvement with +5% gains in 12/19 participants.” Idem, what about the 7 other participants? 

5 participants had a grip strength change of <5%, 2 participants did not complete enough assessments to calculate grip strength or range of motion during the intervention (see Table 3 under the section: Results, Scientific feasibility). 

To provide additional context, the discussion has been changed to the following: 

“Accordingly, we saw 18% average improvement in non-dominant grip strength relative to the dominant side across participants. However, normative data for children’s grip strength, as compared to the dominant side using the modified sphygmomanometer are not available”. – Section: Discussion, Body Function

Further, we have added the following limitation: 

“Next, level-change groupings (small, moderate, and large) for active wrist extension were based on a minimal detectible difference of five degrees [42]. However, similar level-change groupings in grip strength were undeterminable since minimal detectible differences for grip strength, as compared to the dominant hand, do not yet exist for young people with CP. Scores are reported as a percent of the dominant side since improved capacity for bimanual activities is a primary goal for many young people with CP. For context, raw score increases in the affected hand's grip strength averaged 17 mmHg and all but two participants saw an improvement of greater than 10 mmHg. A minimal detectible difference of seven mmHg and a within-subject Standard Error of Measurement of three mmHg has recently been reported in individuals with Parkinson's Disease using a similar modified sphygmomanometer test [71]. These visual analyses summaries are not provided as a definitive evaluation but to aid the reader in their interpretation of the effect size in this single-case design intervention [23].”– Section: Discussion, Limitations

11 Since it is a feasibility study the author should discuss the feasibility of implementing such kind of interventions in “real life” conditions. Do they think it is possible to have the pre-session and the different control sessions organized in practice (and who is going to pay for that). This is particularly important since apparently only 2 of the 19 participants have physiotherapy sessions (so basically no intervention). 

The reviewer raises an important concern. We have added the following text to address the reviewers question surrounding ‘real life’ implementation: 

“It is important to note that this is the first investigation of the novel home-based gaming intervention and accordingly focused on process and scientific feasibility. Specifically, the study focuses on determining the extent to which the intervention could be used at home and estimating the effect size and variance it might have. Towards conducting a complete RCT, more studies and development would be needed to ascertain real-world feasibility, addressing resources and management. This includes a proper health economics analysis and evaluation of the logistic organization with respect to an institute’s existing therapeutic practices.” – Discussion, Implementation. 

Main points related to process feasibility are in the discussion under the sections: Recruitment and adherence, Motivation and Limitations. To further comment on scientific feasibility, as described by Thabane et al 2010, we discuss Body Functions and Activity and Participation outcomes in the discussion as well. 

 

References

Thabane L, Ma J, Chu R, Cheng J, Ismaila A, Rios LP, et al. A tutorial on pilot studies: the what, why and how. BMC Med Res Methodol. 2010;10: 1.

Eldridge SM, Chan CL, Campbell MJ, Bond CM, Hopewell S, Thabane L, et al. CONSORT 2010 statement: extension to randomised pilot and feasibility trials. Pilot Feasibility Stud. 2016;2: 64. doi:10.1186/s40814-016-0105-8

Manolov R, Gast DL, Perdices M, Evans JJ. Single-case experimental designs: Reflections on conduct and analysis. Neuropsychol Rehabil. 2014;24: 634–660. doi:10.1080/09602011.2014.903199

Please see Appendix A (Response to Reviewers.docx) for Screenshots of ClinicalTrials.gov showing trial registration verification date before recruitment began.

---

## [Decision Letter · Decision Letter 1]

15 May 2020

PONE-D-19-22575R1

A biofeedback-enhanced therapeutic exercise video game intervention for young people with cerebral palsy: A randomized single-case experimental design feasibility study.

PLOS ONE

Dear Mr. MacIntosh,

Thank you for submitting your manuscript to PLOS ONE. After careful consideration, we feel that it has merit but does not fully meet PLOS ONE’s publication criteria as it currently stands. Therefore, we invite you to submit a revised version of the manuscript that addresses the points raised during the review process.

Please address the minor corrections requested by reviewer 3 and check syntax and grammar throughout the text. 

We would appreciate receiving your revised manuscript by May 31st. To enhance the reproducibility of your results, we recommend that if applicable you deposit your laboratory protocols in protocols.io, where a protocol can be assigned its own identifier (DOI) such that it can be cited independently in the future. For instructions see: http://journals.plos.org/plosone/s/submission-guidelines#loc-laboratory-protocols

We look forward to receiving your revised manuscript.

Kind regards,

Andrea Martinuzzi

Academic Editor

PLOS ONE

Reviewers' comments:

Reviewer's Responses to Questions

**Comments to the Author**

1. If the authors have adequately addressed your comments raised in a previous round of review and you feel that this manuscript is now acceptable for publication, you may indicate that here to bypass the “Comments to the Author” section, enter your conflict of interest statement in the “Confidential to Editor” section, and submit your "Accept" recommendation.

Reviewer #3: (No Response)

2. Is the manuscript technically sound, and do the data support the conclusions?

Reviewer #3: Yes

3. Has the statistical analysis been performed appropriately and rigorously? 

Reviewer #3: N/A

4. Have the authors made all data underlying the findings in their manuscript fully available?

Reviewer #3: Yes

5. Is the manuscript presented in an intelligible fashion and written in standard English?

Reviewer #3: Yes

6. Review Comments to the Author

Reviewer #3: This is a very nicely done, and well-reported study, on an important topic. I only have a few comments; overall, I enjoyed reading the paper and commend the authors on their clear presentation of the results.

l.72 "participants'" not "participant's" The plural is implied later in the sentence with pronoun "their." Throughout the paper, the authors should check pronoun agreement between subject and verb. In a number of places, a singular subject is followed by a plural pronoun; e.g., he...their (sorry, ex-grammar teacher).

l. 145. Why is "Normal or corrected-to-normal vision and hearing" an EXCLUSION criteria? Is that correct?

In several places, the authors give an abbreviation, followed by the full name that is being abbreviated in parentheses (ll.100-01, ll.164-65, ll.170-71). This order should be reversed (as it is elsewhere in the paper).

Table 3 summarizes Reps, Minutes Active, and Minute in System using means and standard deviations. Judging by the size of the standard deviations relative to the means, these do not appear to be normally distributed, and would be better summarized with medians and interquartile ranges to give a more accurate picture of the data distributions.

25% of the participants did not reach their goals due to a loss of interest. This seems fairly high. Did the authors factor this into their assessment of feasibility? I know they talk about the need to vary programs for different ages and over time to avoid this, but is THAT feasible? I.e., how much money and time go into designing and implementing a single game?

7. PLOS authors have the option to publish the peer review history of their article (what does this mean?). If published, this will include your full peer review and any attached files.

Reviewer #3: No

---

## [Author Response · Author response to Decision Letter 1]

17 May 2020

1 This is a very nicely done, and well-reported study, on an important topic. I only have a few comments; overall, I enjoyed reading the paper and commend the authors on their clear presentation of the results.

 Thank you for your comments we appreciate your suggestions and the time spent reviewing.

2 l.72 "participants'" not "participant's" The plural is implied later in the sentence with pronoun "their." Throughout the paper, the authors should check pronoun agreement between subject and verb. In a number of places, a singular subject is followed by a plural pronoun; e.g., he...their (sorry, ex-grammar teacher).

 Thank you for this comment since English language grammar is not a particular strength of our group. The manuscript has been reviewed for pronoun agreement and the corrections have been made as needed, including editing l.72 from “participant’s” to “participants”.

3 l. 145. Why is "Normal or corrected-to-normal vision and hearing" an EXCLUSION criteria? Is that correct?

 This is indeed a typo. That item has been moved from the exclusions to the inclusions as it should be. Thank you for noticing the mistake.

4 In several places, the authors give an abbreviation, followed by the full name that is being abbreviated in parentheses (ll.100-01, ll.164-65, ll.170-71). This order should be reversed (as it is elsewhere in the paper).

 The location of abbreviations have been corrected to appear in brackets after the full name at each of the lines identified by the reviewer.

5 Table 3 summarizes Reps, Minutes Active, and Minute in System using means and standard deviations. Judging by the size of the standard deviations relative to the means, these do not appear to be normally distributed, and would be better summarized with medians and interquartile ranges to give a more accurate picture of the data distributions.

 Table 3 has been updated to include median and interquartile range as the reviewer suggests giving a more accurate picture of the data distributions.

6 25% of the participants did not reach their goals due to a loss of interest. This seems fairly high. Did the authors factor this into their assessment of feasibility? I know they talk about the need to vary programs for different ages and over time to avoid this, but is THAT feasible? I.e., how much money and time go into designing and implementing a single game?

 The reviewer addresses an important aspect here. Adherence is a central issue. In the current study we found that adherence was relatively high compared to other studies which have shown participants complete 53-78% of practice goals, as compared to 75% in the current work. (l.381-383). From a feasibility perspective, the reviewer also raises a salient issue. The cost of activity development relative to the expected use and benefit will dictate quantity and quality. We mitigated some of this cost and improved feasibility by finding a suitable game and subsequently collaborating with directly with the developer. We have expanded the limitations section to include this point and it now read as follows: 

“…greater choice and game variety, while challenging to implement in rehabilitation protocols, would help maintain novelty and interest in the activity. Collaboration with independent game developers can improve feasibility by offering content with relatively quick and flexible modification abilities, as was the case in the current study with the adapted commercial video game (Dashy Square). [71]” – l.472.

---

## [Decision Letter · Decision Letter 2]

3 Jun 2020

A biofeedback-enhanced therapeutic exercise video game intervention for young people with cerebral palsy: A randomized single-case experimental design feasibility study.

PONE-D-19-22575R2

Dear Dr. MacIntosh,

We are pleased to inform you that your manuscript has been judged scientifically suitable for publication and will be formally accepted for publication once it complies with all outstanding technical requirements.

With kind regards,

Andrea Martinuzzi

Academic Editor

PLOS ONE

Additional Editor Comments (optional):

Reviewers' comments:

Reviewer's Responses to Questions

**Comments to the Author**

1. If the authors have adequately addressed your comments raised in a previous round of review and you feel that this manuscript is now acceptable for publication, you may indicate that here to bypass the “Comments to the Author” section, enter your conflict of interest statement in the “Confidential to Editor” section, and submit your "Accept" recommendation.

Reviewer #3: All comments have been addressed

2. Is the manuscript technically sound, and do the data support the conclusions?

Reviewer #3: Yes

3. Has the statistical analysis been performed appropriately and rigorously? 

Reviewer #3: Yes

4. Have the authors made all data underlying the findings in their manuscript fully available?

Reviewer #3: Yes

5. Is the manuscript presented in an intelligible fashion and written in standard English?

Reviewer #3: Yes

6. Review Comments to the Author

Reviewer #3: (No Response)

7. PLOS authors have the option to publish the peer review history of their article (what does this mean?). If published, this will include your full peer review and any attached files.

Reviewer #3: No

---

## [Editor Report · Acceptance letter]

10 Jun 2020

PONE-D-19-22575R2 

A biofeedback-enhanced therapeutic exercise video game intervention for young people with cerebral palsy: A randomized single-case experimental design feasibility study. 

Dear Dr. MacIntosh:

I'm pleased to inform you that your manuscript has been deemed suitable for publication in PLOS ONE. Congratulations! Your manuscript is now with our production department. 

Kind regards, 

on behalf of

Dr. Andrea Martinuzzi 

Academic Editor

PLOS ONE